# A Remote Sensing–based Intensity-Duration Threshold, Faifa Mountains, Saudi Arabia

Sita Karki[1], Mohamed Sultan[1], Saleh A. Al-Sefry[2], Hassan M. Alharbi[2], Mustafa Kemal Emil[1], Racha Elkadiri[3], and Emad Abu Alfadail[2]

[1] Department of Geological and Environmental Sciences, Western Michigan University, Kalamazoo, MI 49008, USA
[2] Saudi Geological Survey, Jeddah, Kingdom of Saudi Arabia
[3] Department of Geosciences, Middle Tennessee State University, Murfreesboro, TN 37132, USA

*Correspondence to*: Mohamed Sultan (mohamed.sultan@wmich.edu)

**Abstract.** Construction of intensity-duration (ID) thresholds and early warning and nowcasting systems for landslides (EWNSL) are hampered by the paucity of temporal and spatial archival data. This work represents significant steps towards the development of a prototype EWNSL to forecast and nowcast landslides over Faifa Mountains in the Red Sea Hills. The developed methodologies rely on readily available, temporal, archival Google Earth and Sentinel-1A imagery, precipitation measurements, and limited field data to construct an ID threshold for Faifa. The adopted procedures entail the generation of an ID threshold to identify the intensity and duration of precipitation events that cause landslides in the Faifa Mountains, and the generation of pixel-based ID curves to identify locations where movement is likely to occur. Spectral and morphologic variations in temporal Google Earth imagery following precipitation events were used to identify landslide-producing storms and generate the Faifa ID threshold ($I = 4.89D^{-0.65}$). Backscatter coefficient variations in radar imagery were used to generate pixel-based ID curves and identify locations where mass movement is likely to occur following landslide-producing storms. These methodologies accurately distinguished landslide-producing storms from non–landslide producing ones and identified the locations of these landslides with an accuracy of 60%.

## 1 Introduction

Mountainous areas worldwide with steep slopes, high precipitation, and limited vegetative cover often experience landslides. Two main types of landslides are often reported from Faifa Mountains in Saudi Arabia. The first are debris flows that occur when water-saturated soils (largely from weathered bedrock and fragmented rock) move down mountainsides, get channelled into streams, pick up objects along their paths, and deposit their thick load down the valley slopes (Iverson, 1997). The second type results from failure along preexisting fracture planes (Lowell, 1990) that occurs when the following conditions are met (Norrish and Wyllie, 1996): (1) the strike of the planar discontinuity is similar (within 20°) to the strike of the slope face; (2) the dip of the planar discontinuity is less than that of the slope face and oriented in the same general direction; (3) the dip of the planar discontinuity is greater than the angle of the friction of the surface; and (4) the friction angle of the rock material is partially controlled by the size and shape of the grains exposed on the fracture surface and by the mass of the block above the

planar discontinuity (Alharbi et al., 2014). These two types of landslides pose a substantial threat to human life and property in mountainous areas, especially in populated regions that are witnessing unplanned urbanization.

One such area is the Faifa Mountains (area: 119 km$^2$) in the Jazan Province of Saudi Arabia (Fig. 1a). The Faifa area has high population density (~35,000 inhabitants in 137 km$^2$; MMRA, 2017), receives relatively high precipitation (mean annual precipitation [MAP]: 252 mm/yr; Fig. 1b) compared to the remaining parts of Saudi Arabia (83 mm/yr), has steep slopes (up to 65°; Fig. 2b), and witnesses frequent landslide occurrences (1–4 events/year during the study period [2007 to 2017]). Unfortunately, monitoring programs for landslide occurrences (date and time) and conditions (intensity and duration of precipitation) are very limited in Faifa. If such programs existed, they would have generated the archival landslide data needed for the construction of intensity-duration (ID) relationships and for the development of an early warning and nowcasting system for landslides (EWNSL). The paucity of such data in Faifa is largely caused by its rugged nature and the limited coverage of its road network; both factors render many areas inaccessible and hinder the development of monitoring systems. Despite the absence of organized monitoring programs for landslides in the study area, a few were recorded by the Saudi Geological Survey (SGS) in the past few years. Our field observations, and those collected by the SGS in the study area and in its surroundings (Youssef et al., 2014), revealed that debris flows are by far the most prominent landslide type in the study area. To compensate for the deficiencies in field data in Faifa, we complemented the available field data with observations extracted from readily available temporal remote sensing data. These include high-resolution Google Earth images (spatial resolution: 15 m to 15 cm), Sentinel-1A Ground Range Detected (GRD level-1 Synthetic Aperture Radar [SAR] data; spatial resolution: 10 m), satellite-based precipitation data including Tropical Rainfall Measuring Mission (TRMM, 3-hourly_3B42 v7, spatial resolution: 0.25° × 0.25°, ~ 30 km in Faifa) and Global Precipitation Measurement (GPM) IMERG Final Precipitation L3 Half Hourly (V05, spatial resolution: 0.1° × 0.1°, ~12 km in Faifa), and a high-resolution digital elevation model (TanDEM-X DEM; spatial resolution: 12 m). Although GPM provided higher temporal resolution (half-hourly) data compared to TRMM (3-hourly) data, both sensors provided rainfall intensity data in mm/h allowing continuous measurements of rainfall with consistent observational parameters and acceptable (~67%) to high (87%) correlation at the pixel and basin scale respectively (Tang et al., 2016). Field observations were collected (from 2014 to 2016) from the areas that witnessed landslides following precipitation events. In this manuscript, we develop procedures that take advantage of both readily available remotely acquired data and limited field data to develop ID thresholds for the study area, procedures that could potentially be applied to similar areas with limited field data.

A recent review by Segoni et al. (2018a) shows that the majority of the ID-threshold development studies were conducted over well-monitored areas where landslide-related archival data are available from reports, surveys, fieldwork sets (e.g., Burtin et al., 2009; Erener and Düzgün, 2013; Staley et al., 2013; Lagomarsino et al., 2015; Vallet et al., 2016; Piciullo et al., 2017), or even from automatic systems (Battistini et al., 2017). In these areas, several advanced thresholding techniques (e.g., geotechnical process-based, empirical, and rainfall measurement-based) are common (Guzzetti et al., 2007). Unfortunately,

Faifa lacks the historical landslides archives and rain gauge measurements to implement any of these thresholding techniques. Using observation from limited field data and satellite-based data sets (e.g., intensity and duration of precipitation, or location of landslides), we developed rainfall-based ID thresholds. In this respect, our approach does not require extensive archival field data sets to generate ID thresholds. Thus, the approach could potentially be applied in many of the world's mountainous
locations lacking adequate archival field data.

The majority of the ID thresholds that were constructed for various landslide types (e.g., shallow landslides and debris flow [Caine, 1980; Innes, 1983, Crosta and Frattini, 2001; Aleotti, 2004; Jakob et al., 2012]; soil slips [Clarizia et el., 1996], and postfire debris flow [Cannon and Gartner, 2005; Cannon et al., 2011]) provide the magnitude and intensity of rainfall that
triggers landslides but not the locations where they are likely to occur. We generated a unified ID threshold for the Faifa Mountains to identify the landslide-triggering precipitation events and pixel-based thresholds to identify locations where landslides are likely to occur. The pixel-based threshold is adjusted to the response of the individual pixels to historical rainfall events.

Several advances in rainfall thresholding techniques were developed to account for the role of antecedent rainfall conditions preceding landslide development (e.g., Kim et al., 2014, Hong et al., 2017). Others consider software applications that rely on rain gauge records, extensive historical data catalogue, or advanced statistical analyses (e.g., Lagomarsino et al., 2015; Peruccacci et al., 2017; Rossi et al., 2017b). Such techniques cannot be applied in the Faifa area due to the absence of such measurements. Instead we adopt the minimum thresholding technique that was successfully applied in several studies (e.g.,
Caine, 1980; Larsen and Simon, 1993; Cannon et al., 2008; Brunetti et al, 2010; Berti et al., 2012). We acknowledge that if and once such data sets become available for Faifa, the ID thresholds need to be updated to enhance their performance (Rosi et al., 2015).

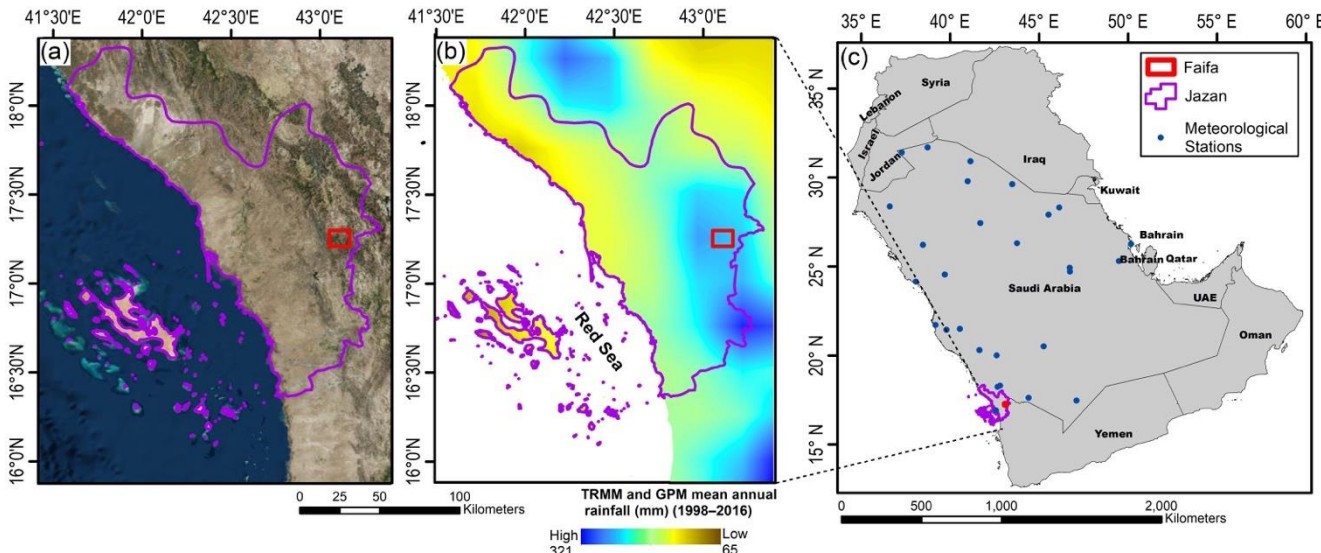

**Figure 1. Location of the study area. (a) Faifa Mountains within the Jazan province. (b) Mean annual precipitation (MAP; 1998–2016) extracted from TRMM (1998–2014, 3B43 level-3, monthly, spatial resolution: 0.25° × 0.25°) and GPM (2014–2016, IMERG level-3, monthly, spatial resolution: 0.1° × 0.1°) showing the higher regional rainfall around Faifa Mountains (MAP in Faifa: 252**
**mm/year) in the southeast part of the Jazan Province. (c) Distribution map of the meteorological stations in the Arabian Peninsula (Mashat and Basset, 2011) in Saudi Arabia.**

## 2 Study area

The study area (119 km$^2$) lies within the Red Sea Hills and covers an area (17.20° N to 17.29° N and from 43.05° E to 43.16° E) proximal to the Saudi-Yemeni border (Figs. 1a and 1c). The elevation is high (rangeing from 259 to 1817 meters above
mean sea level [m.a.m.s.l]) compared to the surrounding lowlands, the topography is steep (slopes as high as 67°; Figs. 2a and 2b), and vegetation is extensive over the mountains but sparse in the surrounding lowlands, as shown in the normalized difference vegetation index (NDVI) map (Fig. 2c). The Faifa region is located within the north-to-northeast trending Tayyah tectonic belt that consists of a complex of metamorphosed volcanic and pyroclastic rocks of basaltic, andesitic, and clastic metasedimentary rocks (Greenwood et al., 1983) that were generated and accreted in an island arc setting some 800 to 900 Ma
(Stoeser and Camp, 1985). The Faifa Mountains are predominantly composed of highly foliated, deformed, and weathered rocks of variable compositions including granite gneiss, amphibolite schist, phyllite, quartzite, biotite, and sericitic schist that are intruded by a massive intergranular syenite (Schmidt et al., 1973; Greenwood, 1979; Greenwood et al., 1983; Alharbi et al., 2014). The area is highly dissected by north-south, northwest-southeast, and east-west trending fault and fracture systems (Fairer, 1985; Alharbi et al., 2014; Fig. 2d). The presence of highly weathered, foliated, and deformed rocks, together with the
high elevations, steep slopes, and sparse vegetation, makes this area prone to landslides even under modest precipitation intensities.

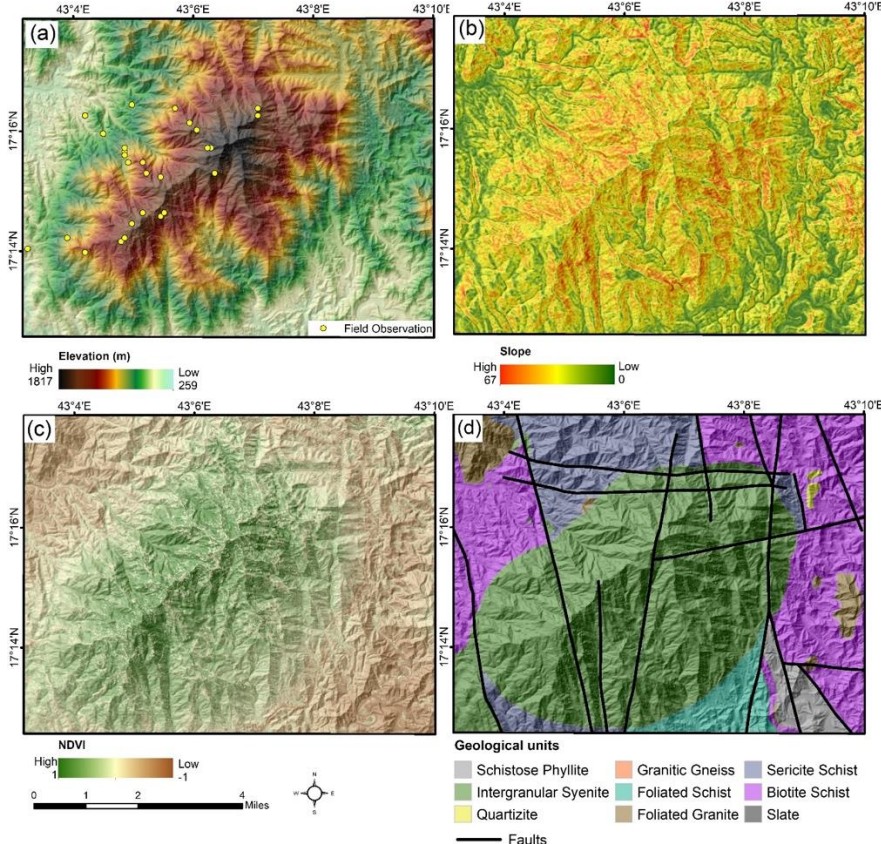

**Figure 2. Maps showing the study area. (a) Elevation map showing locations where field observations were collected for the period extending from February 26 to March 7, 2016. (b) Slope map generated from TanDEM-X DEM. (c) NDVI map generated from Sentinel-2 data (date of acquisition: 2017-06-05). (d) Geologic map for the Faifa Mountains (after Fairer, 1985 and Alharbi et al., 2014).**

Analysis of TRMM (1998–2014) and GPM (2014–2016) measurements for the period 1998 to 2016 revealed sparse precipitation over the Saudi Arabian landscape (MAP: 83 mm/year) but relatively higher precipitation over the Red Sea Hills in western Saudi Arabia, which receive a higher MAP of 108 mm/year. These analyses involved spectral resampling of GPM data to match the TRMM resolution followed by zonal and pixelwise averaging over the indicated time span. Comparison of TRMM to the resampled and averaged GPM measurements over the study area revealed highly correlated (>85%) values during the period of overlap (March to September 2014). A progressive increase in overall rainfall over Faifa was noted over the past six years (MAP: 2010–2016: 315 mm/year; 1998–2009: 227 mm/year) with the wettest year on record in 2016 (total annual rainfall: 450 mm). Two systems of wind regimes are responsible for the rainfall over Faifa: (1) northerly and northwesterly winter cyclonic regimes from the Mediterranean, and (2) summer monsoons from the Arabian Gulf and the Indian Ocean (Alsharhan et al., 2001).

## 3 Methods

The methodology we developed entailed two main steps. The ID curve for the Faifa Mountains was first generated to identify storms that caused landslides (temporal analysis; Sect. 32.1), and then pixel-based ID curves were constructed to identify the locations where movement is likely to occur (spatial analysis, Sect. 32.2–32.4). The latter step involved: (1) selection, calibration, and pre-processing of radar images; (2) generation of backscatter coefficient difference images as a measure of surface roughness change due to precipitation-induced landslides; and (3) development, refinement, and validation of the model to identify pixels susceptible to movement under user-defined precipitation conditions.

Data used for the study include: (1) temporal Google Earth imagery, (2) Sentinel-1A radar imagery; (3) TRMM and GPM; (4) field observation of landslide locations (debris flow and failure along fracture plane); and (5) the TanDEM-X DEM. A detailed flow chart is presented in Figure 3.

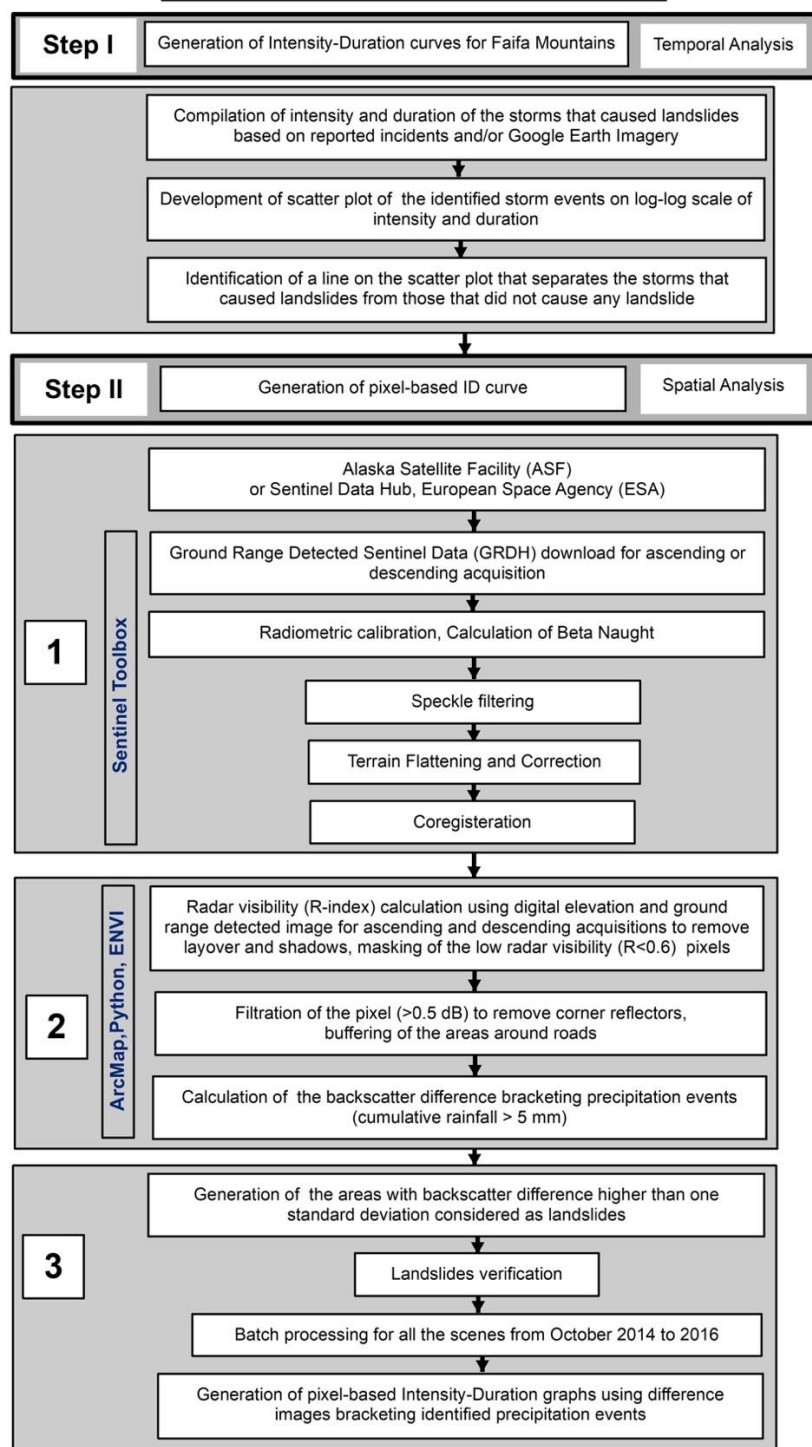

**Early Warning System Development**

**Step I** | Generation of Intensity-Duration curves for Faifa Mountains | Temporal Analysis

Compilation of intensity and duration of the storms that caused landslides based on reported incidents and/or Google Earth Imagery

Development of scatter plot of the identified storm events on log-log scale of intensity and duration

Identification of a line on the scatter plot that separates the storms that caused landslides from those that did not cause any landslide

**Step II** | Generation of pixel-based ID curve | Spatial Analysis

**1** — Sentinel Toolbox

Alaska Satellite Facility (ASF) or Sentinel Data Hub, European Space Agency (ESA)

Ground Range Detected Sentinel Data (GRDH) download for ascending or descending acquisition

Radiometric calibration, Calculation of Beta Naught

Speckle filtering

Terrain Flattening and Correction

Coregisteration

**2** — ArcMap, Python, ENVI

Radar visibility (R-index) calculation using digital elevation and ground range detected image for ascending and descending acquisitions to remove layover and shadows, masking of the low radar visibility (R<0.6) pixels

Filtration of the pixel (>0.5 dB) to remove corner reflectors, buffering of the areas around roads

Calculation of the backscatter difference bracketing precipitation events (cumulative rainfall > 5 mm)

Generation of the areas with backscatter difference higher than one standard deviation considered as landslides

Landslides verification

Batch processing for all the scenes from October 2014 to 2016

Generation of pixel-based Intensity-Duration graphs using difference images bracketing identified precipitation events

**Figure 3.** Flow chart summarizing the developed methodology that could serve as important steps towards the construction of an EWNSL. The developed procedures involved the analysis of temporal Google Earth images, Sentinel-1A radar scenes, and TRMM and GPM rainfall data. Analysis involved two main steps: generation of an ID curve for the Faifa Mountains to identify storms that are likely to produce landslides (landslide-producing storms), and generation of pixel-based ID curves to identify the locations where movement is likely to occur during landslide-producing storms. Step II involved: (1) selection, calibration, and pre-processing of radar images, (2) generation of backscatter coefficient difference images as a measure of surface roughness change due to precipitation-induced landslides and, (3) development, refinement, and validation of the model to identify pixels susceptible to movement under user-defined precipitation conditions. The downloaded scenes were processed using the ESA's Sentinel Toolbox software, ENVI, ArcMap, and Python.

## 3.1 Generation of the ID threshold for the Faifa Mountains

Unfortunately, the distribution of rain gauges is inadequate in the study area (Fig. 1c). There is only one station within Jazan province, and three more stations in its surroundings (Fig. 1c). We utilized the GPM half-hourly (spatial resolution: $0.1° \times 0.1°$) and TRMM 3-hourly (spatial resolution: $0.25° \times 0.25°$) data to extract the intensity and duration of rainfall that caused landslides throughout the period 2007 to 2016 (Table 1 and Fig. 4). In generating the ID threshold for Faifa, we used the peak intensity values; in other words, the shorter sections of the precipitation event with higher intensity were selected. Landslides triggered by the same storm at different locations were assigned the same intensity and duration values. Although semiautomated procedures have been used successfully to extract intensity and duration of landslide-producing precipitation events (e.g., Segoni et al., 2014; Rosi et al., 2016), we adopted a manual approach given the coarse spatial and temporal resolution of satellite data and the limited inventory of historical landslide data over Faifa Mountains. Rainfall events of less than 1 mm/h were omitted given that no landslides were reported from the area at these low rainfall rates and TRMM could mistakenly identify fog for a low rainfall event (<1 mm/h; Milewski et al., 2009). Altogether, 131 precipitation events were extracted from TRMM and GPM data throughout this period, of which 19 events were identified as landslide-producing storms (Table 1). These storms were identified using spectral and morphologic variations associated with landslide development, variations detected in the field and/or extracted visually from pairs of Google Earth images bracketing large precipitation events. Google Earth images were favored over other readily available visible near-infrared (VNIR) satellite data sets given their high spatial resolution (15 m to 15 cm) and long temporal coverage for the study area (2007 to present). An area that witnessed landslides will be covered by spectrally dark vegetation on the Google Earth image preceding the landslide and by spectrally bright rocks and sediments on the image acquired after the landslide development. In many cases the latter image, not the former image, shows a major scar in the source area (onset of the landslide) that gives way to more linear scars in vegetation along the landslide path. It is worth noting that different parts of the Faifa Mountains have differing numbers of Google Earth image acquisitions, amount of coverages, and resolutions.

Starting in October 2014, the SGS initiated a program to field-verify reports of landslide occurrences. Field observations were conducted by our research team following the December 25, 2015 landslide-producing storm and by the SGS researchers throughout the period from October 2014 to October 2016. Our collective field investigations revealed extensive landslides following the events on December 25, 2015, April 13, 2016, April 29, 2016, August 1, 2016, and August 25, 2016. During a

number of these storms, landslides were reported from the mountainous areas proximal to, but outside of, the study area (e.g., Youssef et al., 2014).

Using detected storm-induced spectral and morphologic variations in pairs of archival temporal Google Earth imagery, two additional landslide-producing events (November 18 and December 1, 2014) were extracted; these were apparently not

reported to, or verified by, the SGS researchers during their field campaign due to their location in inaccessible areas. Using the same techniques (storm-induced spectral and morphologic variations) 16 storms were detected in the period (2007 through 2014) preceding the SGS field campaign (2014–2016). Given the paucity of Google Earth images (18 images in 10 years), a number of precipitation events are likely to have occurred between consecutive Google Earth images. If landslides were detected within the period covered by the consecutive Google Earth image acquisitions, it was assumed that the largest of these

storms caused the observed landslides. In the case of some of the identified precipitation events, the rainfall intensity and duration varied from one part of Faifa to another which resulted into the inclusion of more than one landslide-producing storm events between the same set of Google Earth images. In doing so, a few landslide-producing storms and those that did not cause landslides were not identified. The latter type of storms were identified during the field campaign period (Table 1).

**Table 1. Intensity and duration of the precipitation events used for the construction of the Faifa ID curve. Landslide-producing storms were verified through field observations and by examining spectral and morphologic variations in pairs of Google Earth archival images bracketing significant storm events.**

|   | Intensity (mm/h) | Duration (h) | Storm Date | Landslides | Google Earth Imagery Dates |
|---|---|---|---|---|---|
| 1 | 3.17 | 3.00 | 2007-06-02 | Yes | Google Earth (2007-12-30 and 2007-03-01) |
| 2 | 4.83 | 3.00 | 2008-10-11 | Yes | Google Earth (2010-04-19 and 2007-12-30) |
| 3 | 5.34 | 3.00 | 2008-10-24 | Yes | Google Earth (2010-04-19 and 2007-12-30) |
| 4 | 2.58 | 3.00 | 2010-07-29 | Yes | Google Earth (2010-10-28 and 2010-05-10) |
| 5 | 2.69 | 6.00 | 2010-07-11 | Yes | Google Earth (2010-10-28 and 2010-05-10) |
| 6 | 3.62 | 3.00 | 2010-08-25 | Yes | Google Earth (2010-10-28 and 2010-05-10) |
| 7 | 1.85 | 6.00 | 2011-07-31 | Yes | Google Earth (2012-03-05 and 2010-10-28) |
| 8 | 1.86 | 9.00 | 2011-08-27 | Yes | Google Earth (2010-10-28 and 2012-03-05) |
| 9 | 3.29 | 3.00 | 2011-08-28 | Yes | Google Earth (2012-03-05 and 2010-10-28) |

| 10 | 2.91 | 3.00 | 2012-06-21 | Yes | Google Earth (2013-04-14 and 2012-03-05) |
|----|------|------|------------|-----|------------------------------------------|
| 11 | 1.63 | 6.00 | 2013-07-22 | Yes | Google Earth (2013-10-11 and 2013-04-14) |
| 12 | 1.04 | 12.00 | 2014-05-18 | Yes | Google Earth (2014-12-24 and 2014-01-06) |
| 13 | 3.70 | 2.00 | 2014-11-18 | Yes | Google Earth (2014-12-24 and 2014-05-23) |
| 14 | 5.77 | 4.00 | 2014-12-01 | Yes | Google Earth (2014-12-24 and 2014-10-21) |
| 15 | 2.42 | 1.00 | 2015-03-22 | No | Field Visit |
| 16 | 4.67 | 0.50 | 2015-06-02 | No | Field Visit |
| 17 | 1.77 | 0.50 | 2015-06-20 | No | Field Visit |
| 18 | 1.37 | 0.50 | 2015-07-31 | No | Field Visit |
| 19 | 2.39 | 1.00 | 2015-08-25 | No | Field Visit |
| 20 | 3.07 | 1.50 | 2015-09-14 | No | Field Visit |
| 21 | 2.62 | 2.00 | 2015-11-05 | No | Field Visit |
| 22 | 7.91 | 2.00 | 2015-12-25 | Yes | Field visit |
| 23 | 2.85 | 0.50 | 2016-03-25 | No | Field Visit |
| 24 | 5.02 | 6.50 | 2016-04-13 | Yes | Field Visit |
| 25 | 4.76 | 1.50 | 2016-04-29 | Yes | Field Visit |
| 26 | 2.82 | 2.00 | 2016-06-02 | No | Field Visit |
| 27 | 1.64 | 1.00 | 2016-06-15 | No | Field Visit |
| 28 | 8.85 | 12.00 | 2016-08-01 | Yes | Field Visit |
| 29 | 6.40 | 3.00 | 2016-08-25 | Yes | Field Visit |
| 30 | 2.93 | 2.50 | 2016-09-30 | No | Field Visit |

The data presented in Table 1 were plotted to extract the ID threshold for the Faifa Mountains. Landslide-producing storms were represented in Fig. 4 by solid circles (red and black), and the non–landslide producing storms by open circles. The solid black circles are for field-verified landslide-producing storms, and the red circles are for landslide-producing events extracted from Google Earth images. The figure shows the extracted ID curve (red line; equation: $I = 4.89D^{-0.65}$) that provides the best visual separation between the landslide-producing (solid circles above red line) and non-producing (open circles below red line) precipitation events. Given the limited number of storms that were identified throughout the investigated period we believe that the adopted approach for defining the ID threshold is adequate at this stage.

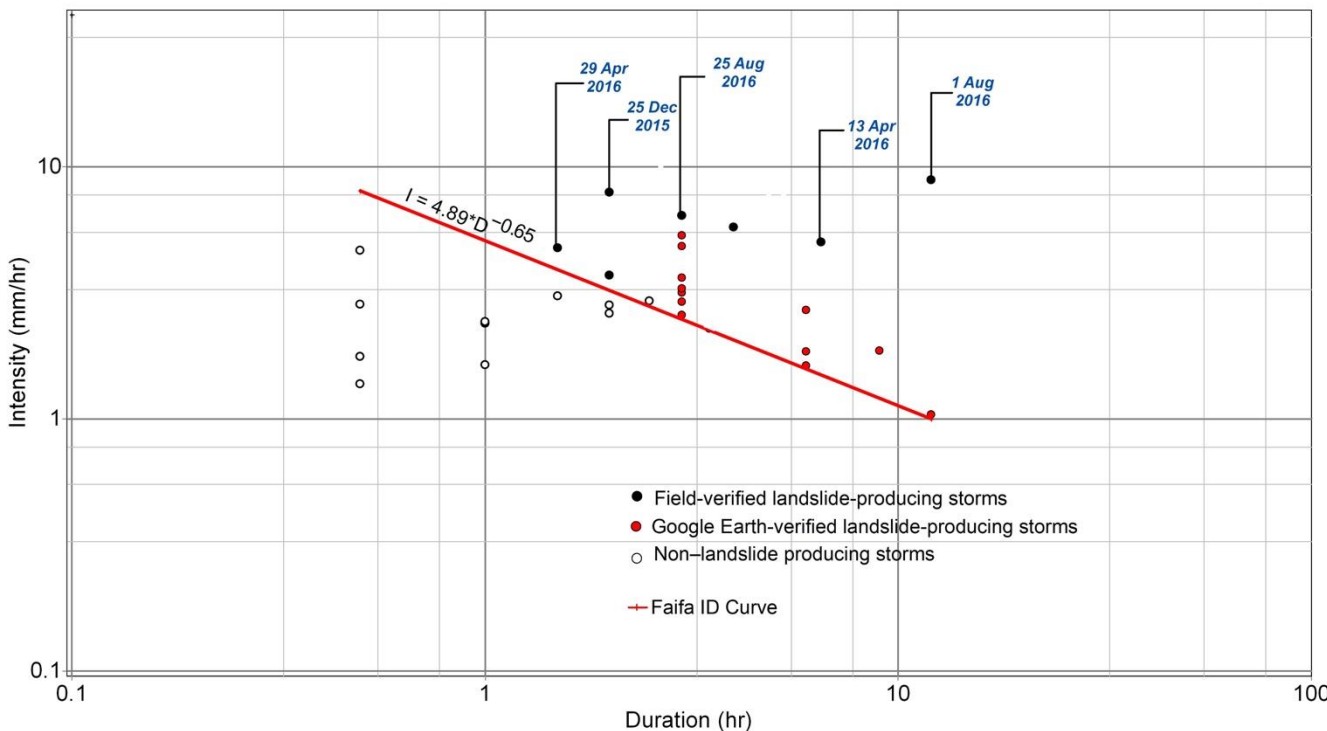

**Figure 4. ID scatter plot generated from landslide-producing storm events (solid circles) and non-producing precipitation events (open circles) during the period 2007 to 2016. The Faifa ID curve (equation: $I = 4.89D^{-0.65}$; duration threshold [x]: 0.5 to 12 h) separates the landslide-producing events from non-producing events.**

## 3.2 Selection, calibration, and pre-processing of radar images

The radar backscatter differences were used to determine the location of the landslide for the storm that caused the landslides. Sentinel-1A radar scenes were downloaded for ascending and descending acquisition modes from the Sentinel Data Hub (https://scihub.copernicus.eu/dhus/#/home), a download platform for the European Space Agency (ESA), for the period between October 2014 and October 2016. The scenes can also be downloaded from the Alaska Satellite Facility's (ASF) website (https://vertex.daac.asf.alaska.edu/). The scenes acquired immediately after (1 day or less) the rainfall were not used in the generation of backscatter coefficient difference images to avoid differences in backscatter due to precipitation-related change in moisture content. The pre-processing steps that were applied to the downloaded scenes included radiometric calibration and calculation of beta naught ($\beta^0$, the radar brightness coefficient), speckle filtering, terrain flattening and correction, and image co-registration. Ascending and descending scenes for the same area provide different degrees of visibility and, depending on the orientation and complexity of the topography (van Zyl et al., 1993), one acquisition mode may provide better visibility than the other.

Standard processing procedures for SAR scenes were applied together with additional filtrations to remove backscatter anomalies that could be confused with our target. Ground range detected (GRD) level 1 images were downloaded and radiometrically calibrated using ESA's Sentinel Toolbox following the basic processing steps established by Veci (2016). The level-1 GRD products are focused SAR data that has been detected, multi-looked, and projected to ground range using an

Earth ellipsoid model (Small and Schubert, 2008). The GRD images were used to calculate the $\beta^0$ (Small, 2011), a measure of radar backscatter energy, in decibels (dB; Raney et al., 1994), for both ascending and descending modes. The existing granular noise that degrades the quality of SAR data, known as speckle, was minimized in the extracted radar backscatter coefficient images using the Lee Filter (window size: $3 \times 3$; Lee, 1983; Lee et al., 2009) and high-resolution DEM (TanDEM-X DEM; resolution: 12.5 m). The Terrain Flattening and the Range Doppler Terrain Correction (Small, 2011) was applied to the speckle

filtered scenes to correct for radiometric biases introduced by the rugged topography of the study area. Each of the processed scenes was co-registered (sub-pixel co-registration) to the previously acquired one in Sentinel Toolbox (Press et al., 1992). Following the generation of the backscatter images, the scenes were cropped to the extent of the Faifa area to facilitate the execution of the steps that follow. The details of the processes have been provided in Fig. 3 (step II).

### 3.3 Generation of backscatter difference images

Following the identification of precipitation events over Faifa, backscatter difference images were generated between scenes bracketing the identified precipitation events. The initial analysis of these difference images revealed that corner reflectors and areas of low visibility can produce a response similar to that of landslides; hence, procedures were developed to identify and mask out these areas. The generation of the backscatter difference images involved a number of steps: (1) calculation of radar visibility and removal of low visibility areas; (2) identification and removal of corner reflectors; and (3) generation of

backscatter difference images.

Ascending and descending scenes for the same area provide different degrees of visibility depending on the topography and satellite orientation. (Notti et al., 2014). A radar visibility index (R; Notti et al., 2014) image was used to identify and mask out areas of low visibility in both the ascending and descending backscatter images. The R index is a function of local variables

(slope, aspect, incidence angle, layover, and shadow), and satellite geometry (line-of-sight azimuth). Using high-resolution DEMs, digital images were computed for each of those variables, which were then used to generate R index images for ascending and descending geometries. These R index products were applicable for all backscatter scenes of same geometry, and its values range from 0 (low visibility) to 1 (high visibility). Pixels with R values below a threshold of 0.6 were found to be spatially correlated with areas affected by overlays and by shadowing and were masked out. The distribution of pixels with

backscatter coefficients exceeding 0.5 dB were found to correlate with that of buildings, construction areas, vehicles, and parking spaces. Such features can act as corner reflectors and produce high radar returns by reflecting waves towards the source. Pixels with backscatter coefficients exceeding 0.5 dB (corner reflectors) were masked out. The filtered backscatter

images were used to generate backscatter difference images between pairs of consecutive backscatter scenes, and only those pairs bracketing storm events were considered for further analysis.

## 3.4 Refinement and validation of the model

The refinement and validation of the model involved: (1) spatial refinement and standard deviation (SD) image generation; (2) field verification; and (3) batch processing of scenes. The population density in and around the road networks is high, and so are the risks for human and property losses if landslides occur in their vicinity. The construction of roads can trigger debris flows, especially in cases when roads intersect steep slopes (Fig. 5a) or terraces constructed on these steep slopes (Fig. 5b), ephemeral valleys, and fracture planes dipping towards the road (Fig. 5c; Alharbi et al., 2014).

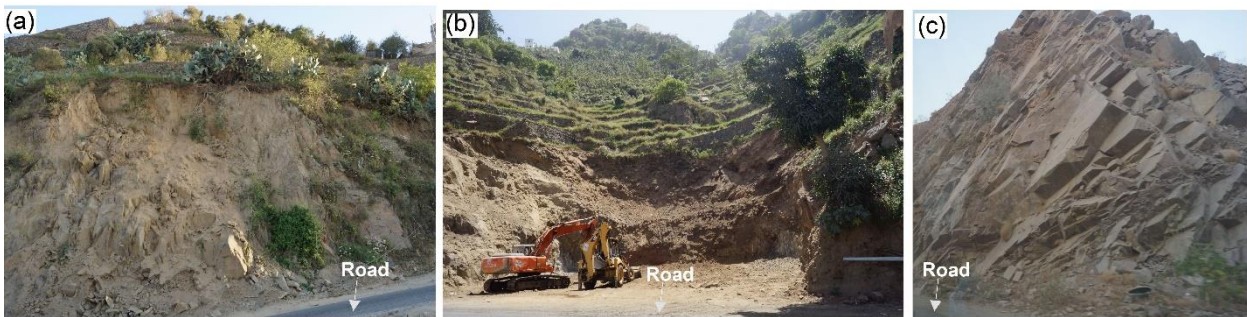

**Figure 5. Landslides proximal to, possibly triggered by, road construction and intensified by rainfall (a) Debris flow caused by failure on steep slopes intersected by roads. (b) Debris flow caused by failure of terraces constructed on steep slopes. (c) Landslide caused by failure on fracture planes dipping towards the road.**

The distribution of historical debris flows in the study area was investigated to identify areas susceptible to debris flow. As described earlier, areas that witnessed recent debris flows are characterized by spectrally bright rocks and sediments, a major scar in the vegetation within the source area (onset of landslide) that gives way to more linear scars in the vegetation along the landslide path. As years go by, spectral and morphologic features indicative of debris flows can get obscured by encroaching vegetation making it more difficult to identify the older debris flows. Many of the historical debris flows were found on steep slopes, along first order streams, above and proximal to the main roads as shown in Figure 6. Using these three criteria, areas susceptible to debris flows were identified by: (1) extracting stream networks using a stream delineation algorithm (Tarboton et al., 1991) in ArcGIS 10.5 over the steep slopes (>30°) and capturing first order streams using a small flow accumulation value (10 pixels); (2) assigning a buffer zone (20 m wide) around the extracted streams to delineate the areas that are likely to be triggered by runoff during and following rainfall events; and (3) assigning a buffer zone (100 m wide) around the roads. The use of the latter criterion allows the identification of areas susceptible to failure along preexisting fractures as well since our field observations showed that the majority of such failures were triggered by road construction. The selected width of the buffer zones was determined by examining the proximity of the historical landslides to roads and extracted streams.

The selection of the buffered zones for further investigation served two purposes: (1) targeting areas of high risk, and (2) capturing the backscatter variations that are related to landslides, variations that could have been confused with those caused by factors other than landslides (e.g., change in vegetation intensity or vegetative cover) if the entire area was considered. Fig. 6 shows several landslides within areas identified as being susceptible to landslides using the three above-mentioned criteria.

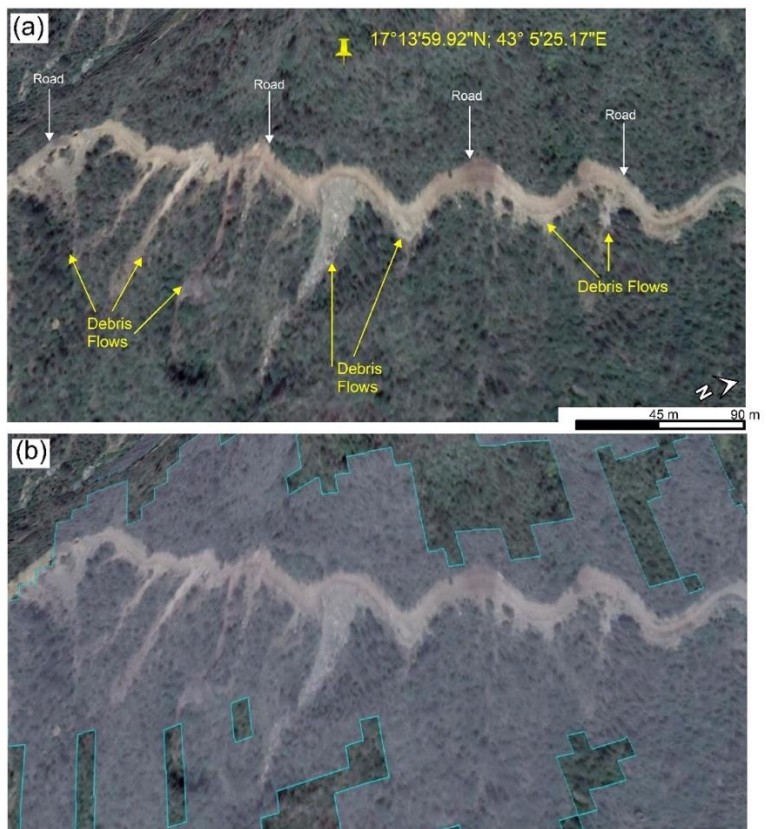

**Figure 6. Google Earth Imagery showing the spatial refinement procedures. (a) Areas showing debris flows within areas characterized by steep slopes (>30º), and proximity to roads (<100 m) and to the first order streams (<20 m). (b) The areas that satisfy these three conditions are outlined by the shaded polygon.**

The spatial refinement was followed by filtration to detect spectral anomalies in the resultant image. A standard deviation–classified image was generated for a backscatter difference image that bracketed the December event (images acquired on December 24, 2015 and February 12, 2016). The differences in backscatter were classified into four groups: area of ≤1SD (no change), >1SD to ≤2SD (lowest change), >2SD to ≤3SD (medium change), and >3SD (highest change). Within the buffered areas on any of the backscatter difference images, the largest variations are expected to correspond to areas that witnessed landslide-related changes in roughness. Field observations following the December 25, 2015 precipitation event (15 mm) were conducted (February 26 to March 7, 2016) to test this assumption. The investigation proved the examination of the variations

in spatially refined and spectrally filtered backscatter difference images and the effectiveness of the applied filtering techniques in omitting the false positives.

Altogether we visited 27 sites in Faifa during our field investigation (Fig. 2a). It was found that the distribution of areas with ≤1SD variations on the extracted difference images did not correspond to any of the observed landslides and are here attributed to temporal variations in vegetation, minor roughness changes, and possibly sub-pixel errors in co-registration. Areas exceeding 1SD on the difference images (Fig. 7; clusters of red, yellow, and green pixels representing highest, medium, and lowest changes, respectively) in the backscatter difference image corresponded to landslide locations and showed evidence for recent redistribution of boulders and sediments in the field. If these conditions were met, a landslide was considered as being verified. Altogether, 90%, 60% and, 86% of the pixels exceeding 3σ, 2σ, and 1σ were located within contiguous areas identified as being locations of landslides. The remaining areas were mostly random distributions of individual pixels resulting from corner reflectors or artifacts due to inadequate speckle filtering. Out of the 15 landslides that were identified with the proposed method, 14 were verified in the field. There was a false positive where the pixels exceeding 1SD corresponded to road construction–related changes (Table 2: site 26). Field investigations of 12 sites verified that spatial refinement and filtration techniques were successful in filtering out 9 of the 12 false positives resulting from corner reflectors (e.g., building, constructions), but mistakenly removed an active debris (Table 2: site 1) and structurally stabilized fracture plane (Table 2: site 19 and 20).

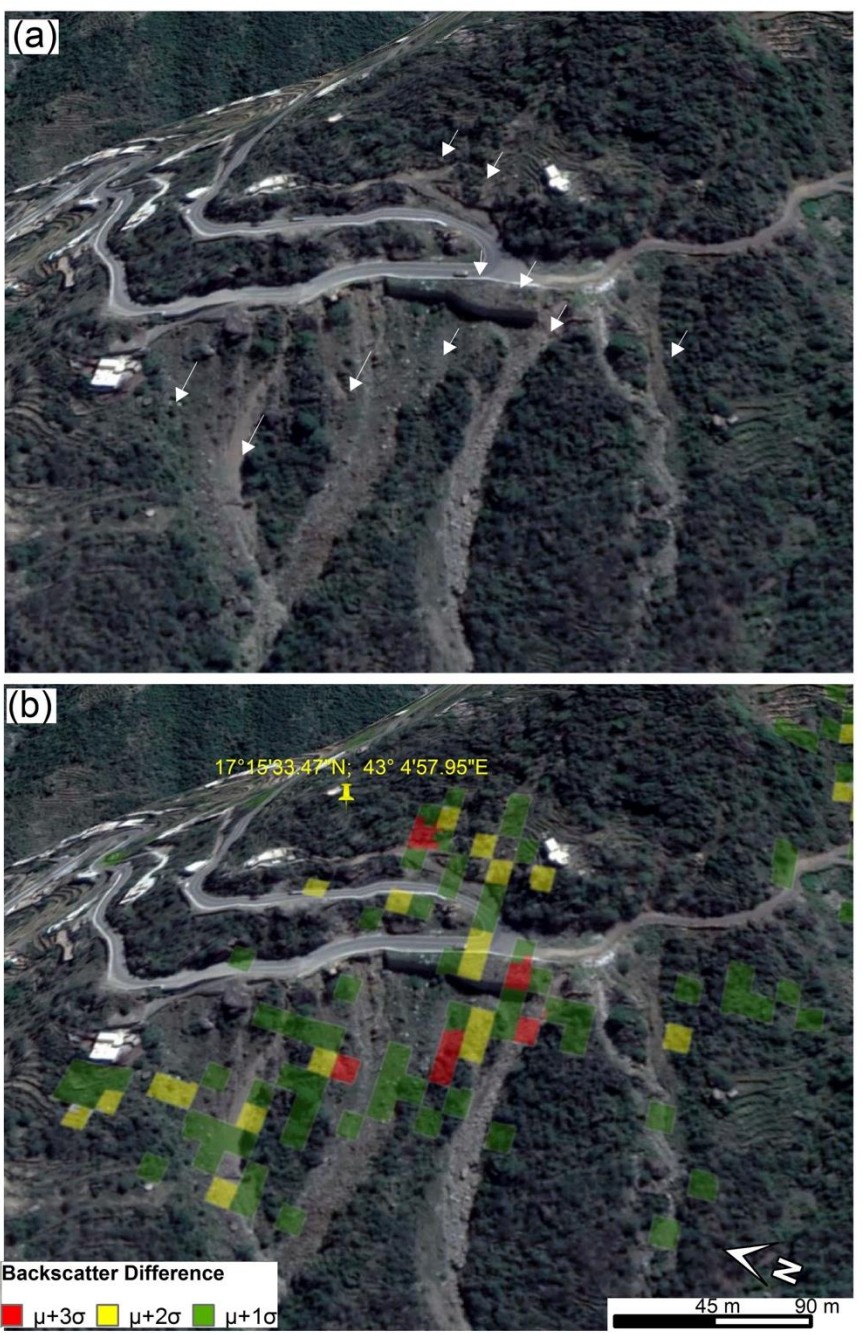

**Figure 7. (a) Google Earth imagery showing the distribution of debris flows (identified by white arrows). (b) Backscatter difference image for two descending scenes bracketing (acquisition dates: December 24, 2015 and February 10, 2016) a precipitation event on December 25, 2015 showing correspondence of areas of low to negligible variations (≤1SD) with vegetation and areas exceeding 1SD (clusters of red, yellow, and green pixels) to debris flow locations that showed evidence for recent redistribution of boulders and cobbles in the field.**

**Table 2. Field observations collected (February 26 to March 7, 2016) for the assessment of radar-based distribution of active landslides, the areas exceeding 1 standard deviation (1SD) on difference images. Locations shown in Fig. 2a.**

| | Difference image | Remarks | Long. (°E) | Lat. (°N) |
|---|---|---|---|---|
| 1 | | *Filtered out active debris flow; false positive* | 43.054 | 17.234 |
| 2 | 3SD, 2SD | Verified active debris flow | 43.065 | 17.237 |
| 3 | 3SD, 2SD | Verified active debris flow | 43.070 | 17.233 |
| 4 | | *Filtered out road construction* | 43.080 | 17.236 |
| 5 | | *Filtered out terraces, bare soil, no vegetation* | 43.081 | 17.237 |
| 6 | 3SD, 2SD | Verified active debris flow, recently mitigated at intersection with road | 43.083 | 17.241 |
| 7 | 2SD | Verified active debris flow | 43.086 | 17.244 |
| 8 | | *Filtered out terraces* | 43.091 | 17.243 |
| 9 | 2SD | Verified active debris flow; locals reported activity during rainfall | 43.092 | 17.244 |
| 10 | 2SD | Verified active debris flow used to dispose construction material | 43.106 | 17.255 |
| 11 | 3SD, 2SD | Verified active debris flow, recently mitigated at intersection with road | 43.105 | 17.262 |
| 12 | | *Filtered out buildings* | 43.118 | 17.271 |
| 13 | 2SD | Verified active debris flow | 43.118 | 17.273 |
| 14 | | *Filtered out road construction* | 43.070 | 17.271 |
| 15 | | *Filtered out road construction* | 43.075 | 17.266 |
| 16 | 2SD | Debris flow related to terraces | 43.081 | 17.261 |
| 17 | 2SD | Verified debris flow recently mitigated proximal to road | 43.081 | 17.262 |
| 18 | 3SD, 2SD | Verified active debris flow | 43.081 | 17.260 |
| 19 | | *Filtered out shotcrete to stabilize the fracture planes; false positive* | 43.087 | 17.255 |
| 20 | | *Filtered out shotcrete to stabilize the fracture planes; false positive* | 43.086 | 17.258 |
| 21 | 3SD, 2SD | Verified active debris flow | 43.082 | 17.258 |
| 22 | 2SD | Verified failure along fracture plane dipping towards the road | 43.091 | 17.254 |
| 23 | | *Filtered out buildings* | 43.104 | 17.262 |
| 24 | | *Filtered out terraces* | 43.101 | 17.267 |
| 25 | 3SD, 2SD | Verified active debris flow bordering a terrace | 43.099 | 17.269 |
| 26 | 3SD, 2SD | Construction related debris flows downhill from the road; false negative | 43.095 | 17.273 |
| 27 | | *Filtered out construction along the road* | 43.083 | 17.274 |

### 3.5 Pixel-based adaptation of Faifa ID threshold as a predictive tool

The ID curve for any pixel should separate landslide-producing events (backscatter difference > 1SD) from non–landslide producing (backscatter difference ≤ 1SD) precipitation events. On these graphs, landslide-producing events plot above the curve, and the non-producing events plot below it.

A pixel-based debris detection system was developed by adopting the slope of the extracted Faifa ID curve. The assignment of the ID curves to the individual pixels will depend on the relative stability of the individual pixel. The less stable pixels, such as those on steep slopes, are expected to experience movement in response to weak, moderate, and extreme storm events, whereas the more stable pixels will move during the extreme events only. Figure 8 showsdemonstrates the ID curve for a more

stable pixel. Curve A represents the ID curve for more stable locations as it showed evidence for landslide-related movement (>1SD on the radar backscatter difference image) in response to five bigger events (December 1, 2014, December 25, 2015, April 13, 2016, August 1, 2016, and August 25, 2016) but no movement (<1SD on the backscatter difference image) following the April 29, 2016 and November 18, 2014 storms. Curve A has the slope of the Faifa ID threshold, but a different intercept. Thus, knowing the historical response of each individual pixel to these storms, each pixel was assigned an ID curve whose

slope is similar to that of the Faifa curve. In other words, the pixel-based ID curve uses the historical landslide response of a pixel to estimate the intensity and duration of the precipitation that would cause landslide in the future. Any event that plots above the pixel ID curve would produce landslides at that location, whereas the one that plots below would not produce a landslide. The pixel-based ID curve helps to predict the location that will witness landslides under any future storm event. With the current limitation of data sets, only few upward translations of Faifa ID curve is possible. As the inventory of landslide

grows, we expect that the placements of pixel-based ID thresholds will get progressively refined and will hence represent more realistic views of the stability of individual pixels. The expanded inventory will also enable the application of advanced thresholding techniques.

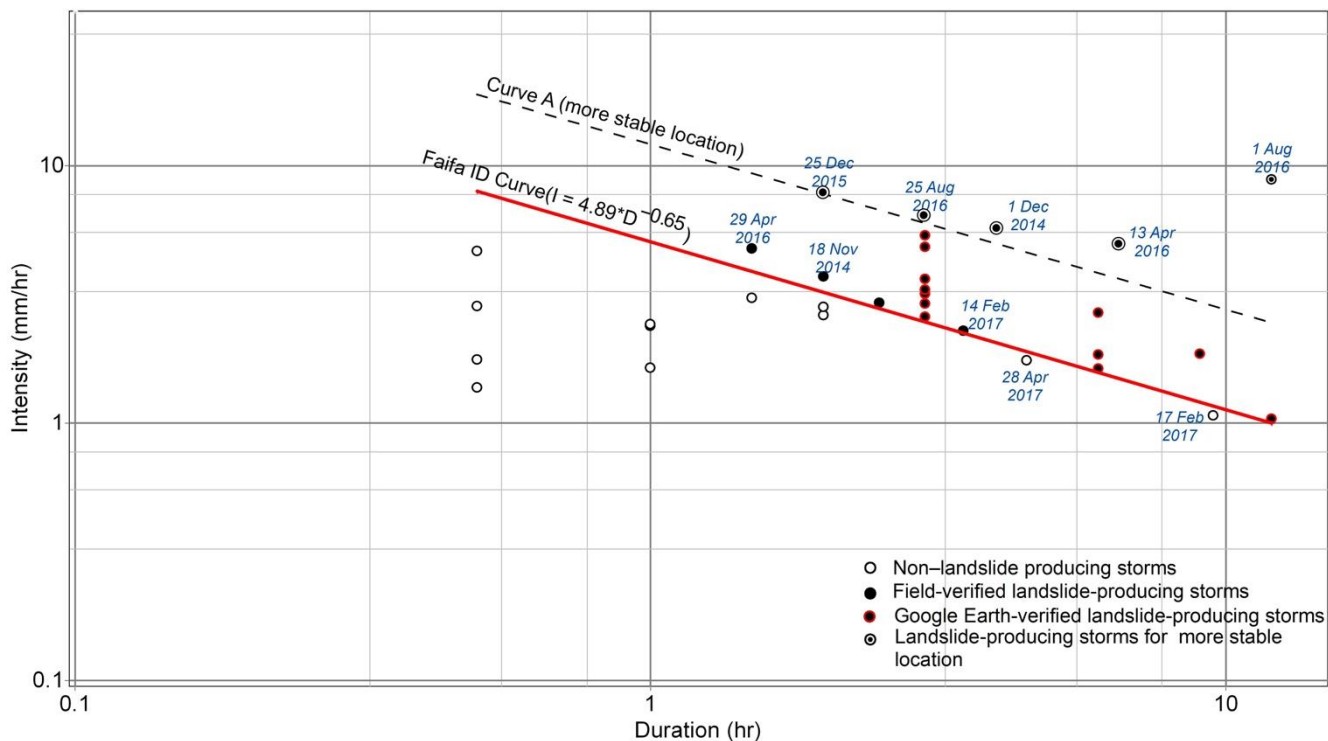

**Figure 8. Demonstration of pixel-based ID curves. Curve A is a curve for a more stable pixel that witnessed landslides in response to five storms (December 1, 2014, December 25, 2015, April 13, 2016, August 1, 2016, and August 25, 2016). The ID curve for the more stable location is parallel to the Faifa ID curve that separates landslide-producing storm events from landslide non-producing storm events from Fig. 4.**

### 3.6 Validation of ID threshold

Three precipitation events larger than the threshold (1 mm/h for 1 h) were recorded during the period from November 2016 to April 2017. These occurred on February 14 (intensity: 2.28 mm/h; duration: 3.5 h), February 17 (intensity: 1.07 mm/h; duration: 9.5 h), and April 28 (intensity: 1.75 mm/h; duration: 4.5 h) of 2017. The event on February 14 plotted above the Faifa ID curve, whereas those on February 17 and April 28 plotted below the curve. Landslides were reported following the February 14 event, but not for the two other storm events, an observation that supports the validity of the extracted ID curve for Faifa.

Using the precipitation intensity and duration for the February 14 storm, and the extracted pixel ID curves, we generated a map showing the areas (three or more pixels) that are likely to witness movement under the specified precipitation conditions. We visually inspected these areas on the Google Earth images that were acquired before (October 2, 2016) and after (March 29, 2017) the February 14 storm as shown in Figure 9. Spectral and morphological variations indicative of landslides were detected on the March 29 Google Earth image. Specifically, 13 landslides were predicted, out of which 6 were verified by

inspecting the March 29 image, an accuracy of 60%. Similarly, out of seven locations where no landslides were predicted, one location witnessed a landslide. We suspect that the high number of false positives (seven locations) is largely an artefact of the adopted method of landslide verification. The morphological variations observed on Google Earth images and indicative of landslides are effective in detecting the large, but not the small, landslides. The higher number of false positives (seven

locations) than the false negative (one location) suggests that the pixel-based ID curve significantly reduces the number of false negatives. With a limited number of post-study storm events, the entire Faifa area currently can accommodate only few adjustments to the ID curve. With the inclusion of more storms and accumulation of archival data, the pixel-based ID curve is expected to represent the unique historical signature of landslide records. Thus, over time, the number of false positives is expected to decrease as more and more areas would have their thresholds based on its relative stability instead of the minimum

threshold established for the Faifa region. At this stage, the reported accuracy is reasonable for developing a prototype EWNSL given (1) the uncertainties associated with extracting the Faifa and pixel-based ID curves, and (2) the fact that landslides in a particular area tend, in some cases, to stabilize the location and reduce the chances of landslide recurrence in the same area.

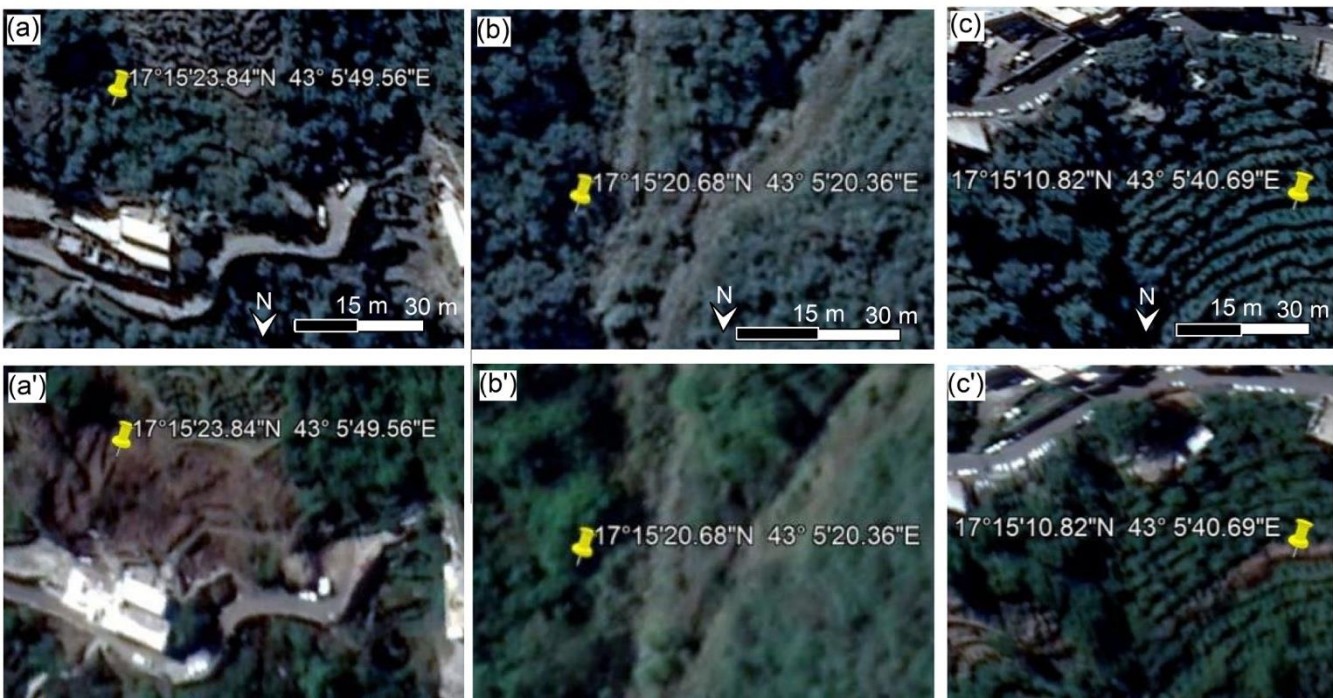

**Figure 9: Demonstration of the prediction result for the storm that occurred on February 14, 2017. (a) Area that**

**witnessed a landslide that was predicted (true positive). (b) Area that did not witness landslide but was predicted (false positive). (e) Area predicted as stable but witnessed landslide (false negative).**

**4 Discussion**

Our ability to predict the landslide-producing storms and the locations of these landslides will depend largely on the accuracy of the extracted/adopted ID curves. The production of the Faifa ID curve was based on precipitation data for 30 storms, approximately 63% of which produced landslides. Precipitation was extracted from the earlier coarse TRMM (3 h; 0.25° × 0.25°) data and later from the finer resolution GPM (1 h; 0.1° × 0.1°) data; field observations and temporal Google Earth images were used to identify which of these storms produced landslides. The temporal coarseness of the precipitation data, especially 3-hourly TRMM data, provides the uncertainty in the precise placement of the ID curve. In upcoming years, additional data points (landslide-producing and non-producing storms), especially those extracted from GPM or rain gauge data with higher spatial and temporal resolution, will be used to refine the initial Faifa ID curve.

Refining the Faifa ID curve will enhance the accuracy of the pixel-based ID curves as well, given that they are assumed to be parallel to the Faifa ID curve. In the construction of these curves, we were constrained by the following limitations in data availability and processing technique: (1) the limited number of storms (18) that occurred throughout the time period (October 30, 2014 to October 31, 2016) during which radar images were available; (2) coarse spatial resolution of the DEM (12.5 m TanDEM-X) and radar data (multi-looked using DEM, 12.5 m) that made it difficult to identify landslides of limited size (<25 m or the size of two pixels); (3) the discontinuous acquisition of Sentinel-1A data (ascending and descending modes) that interrupted the monitoring of landslides in response to storm events; (4) sub-pixel errors in co-registration of radar data and anomalous backscatter spikes originating from buildings and construction activities that produced radar responses similar to landslide-related radar response and were not filtered; (5) drastic changes in the slope and/or vegetation in a particular pixel that impaired the functionality of several pixel-based ID curves; (6) the possibility that frequent rainfall of short duration could have gone undetected given the coarse temporal resolution of the satellite-based precipitation data; and (7) limited field investigations and reliance on Google Earth imagery did not provide enough information to develop a robust thresholding technique.

In coming years, the pixel-based ID curves we developed will be refined by: (1) acquiring high spatial and temporal resolution precipitation data; (2) identifying additional landslide-producing storms to augment the existing database and update the existing pixel-based ID curves; (3) applying additional filtration techniques (e.g., coherence threshold filters to reduce false positives); and (4) developing an urban mask to exclude radar responses from corner reflectors that could be confused with those from landslides. We will also explore refining our methodologies to account for the impact of antecedent precipitation on landslide development (e.g., Chen et al., 2015). To date, the application of ID thresholds for landslide hazard assessment is widespread in early warning systems at local and regional scales (e.g., Peruccacci et al., 2017; Rossi et al., 2017a), yet over the past few years there has been increasing recognition of the role of hydrology in landslide initiation, a factor that is not fully incorporated in the ID threshold analysis. The intensity and duration of rainfall during which a landslide occurs are not the

only triggers for landslides; the rainfall events (antecedent rainfall) that preceded the landslide-causing precipitation are triggers, as well (Kim et al., 2014, Hong et al., 2017). A recent study (Segoni et al., 2018b) highlighted the role of soil moisture content preceding rainfall events in the initiation of landslides and incorporated it in the development of a statistical early warning system. It has been shown that the antecedent and peak rainfall play important roles in triggering landslides in general,

but debris flow development is more related to peak rainfall than antecedent rainfall (Chen et al., 2015). Given that the overwhelming majority of our landslides are debris flows, we do not anticipate that the inclusion of the soil moisture content in our model will largely affect our findings, yet futuristic refinements of the developed methodologies should consider the role of antecedent moisture.

The adopted methodologies and suggested refinements represent significant steps towards the development of a prototype EWNSL. To better achieve this goal, the following additional automated steps have to be accomplished. Near real-time measurements of precipitation should be collected from the rain gauge network over the study area to avoid the delays associated with posting satellite-based precipitation (GPM: 3 to 6 hours). Temporal precipitation distributions can be derived from the acquired rain gauge measurements and used as inputs to our developed modules. Currently, our methodology

identifies vulnerable areas based on user-defined precipitation intensity and duration. Once the nowcasting system is established, as rainfall data is collected, it will be fed automatically into the EWNSL to identify the areas likely to witness landslides at any time. The precipitation at any location could be assumed to continue for a pre-determined time period (e.g., 1 hour) and the model outputs under such assumptions could be used to predict the areas that are likely to witness landslides in that pre-determined time period. The predictive model outputs could be posted in near–real time on a web-based GIS, giving

the authorities and citizens in threated areas enough time to vacate these locations.

**5 Conclusions**

We developed a predictive system that shows whether a storm with a particular intensity and duration can cause landslides in the Faifa Mountains. For the identified landslide-producing storms, the developed methodologies will also select areas that are

likely to witness landslide development. The extracted ID curve for the Faifa is used for the former and the extracted pixel-based ID curves for the latter.

The methodologies advanced here are robust and cost-effective procedures that could be readily applied to many data-deficient locations worldwide. The proposed methodology relies heavily on readily available satellite data and thus could be applicable

to many of the world's mountainous locations. The developed methodologies and rigorous refinements represent significant steps towards the development an EWNSL if precipitation forecasts become available. The proposed procedures for the development of ID curves should not be considered as alternatives to the well-developed field-based ID relationships and to recently introduced advances in such applications but could be used in absence of such field-based datasets.

**Author Contributions:** Sita Karki processed the remote sensing data and prepared the manuscript. Mohamed Sultan supervised the project and helped in the manuscript development. Saleh A. Al-Sefry and Hassan M. Alharbi led the field investigation and data collection. Mustafa Kemal Emil and Racha Elkadiri Racha helped in the radar data processing and statistical analysis. Emad Abu Alfadail provided geographical information system technical support for the project.

**Competing interests:** The authors declare that they have no conflict of interest.

**Acknowledgements:** We would like to thank the Saudi Geological Survey for their scientific, financial, and logistical support for this project. We also acknowledge the European Space Agency for Sentinel-1A data and German Aerospace Center (DLR)

for TanDEM-X data used in this study. We thank two anonymous reviewers for their constructive criticism to bring the manuscript to the present form.

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
