# Peer review of "A Remote Sensing-based Intensity-Duration Threshold, Faifa Mountains, Saudi Arabia"

_Natural Hazards and Earth System Sciences, 2018_

## Referee Comment (RC1) · Anonymous Referee #1 · 18 Dec 2018

Dear authors, I have carefully read your manuscript, and I found it promising. It is well centered on the aims and scopes of the journal, it is written in a good English, and it describes an original approach in a climatic setting that is not usually associated to rainfall induced landslides. However, I identified some shortcomings that need to be fully addressed before publication. Please, take crefully into account my comments below to prepare the revised version of the manuscript. I think that after this round of major revisions, your work could become a good contribution to the journal.

-

GENERAL COMMENTS

1- The abstract is too short. I strongly recommend expanding it, providing more details

on what has been done and the results obtained.

2- In my opinion, the introduction has two main shortcomings and should be reorganized: a) A literature review should be added to describe the existing state of the art and to account for previous works in the field of rainfall thresholds. You can start with some recent reviews (Segoni et al., 2018a; Guzzetti et al., 2008; Guzzetti et al., 2007) and some recent relevant works (Cannon et al., 2011; Jackob et al., 2012; Lagomarsino et al., 2015; Peruccacci et al., 2017; Rossi et al., 2017). Also, NHESS recently published a special issue on a related subject and it could be another starting point. b) The part concerning the description of the case study should be placed into another section devoted to the study area description, merged with the 1.1 section and re-numbered as 2.

3- Some remarks on the methodology. a) What happens when you have two or more landslides during the same storm and they are located in two different pixels? Are they characterized by different I/D values? b) You should better explain if you use the mean rainfall intensity or the peak intensity. In other words: do you consider the total duration of the whole rainfall event or do you identify a shorter section of the event in which you have the higher intensity? c) The comparison with literature threshold is questionable. First, I find little meaning in a graphical comparison among thresholds defined for completely different settings (e.g. the Alps), different processes (e.g. post-fire debris flows), and different background (e.g. Caine used only debris flows with a relevant impact, that's why his global threshold is much higher than yours). Second, it is not clear how you assess which threshold is the most similar to yours. Similar intercept? Similar gradient? To sum up, I suggest either deleting this part or normalizing the threshold equations by the mean annual precipitation (this should allow a straightforward comparison).

4 - The discussion of the results needs to be strengthened. I can suggest some inputs, feel free to explore other directions: a. After your validation, I understand that you have 6 true positives and 7 false positives. How many false negatives? If there is none,

that's something that deserves to be stressed. And these results should be further discussed. E.g. the balance between false positives and false negatives is usually a trade-off that could be modified adjusting the calibration of the curves. b. It would be very interesting for the community working on rainfall thresholds, to highlight what difference in forecasting accuracy is obtained (i) when the only Faifa curve is used as a generic threshold for the whole area; (ii) when your more refined pixel-based approach is used. c. The results shown are maybe too weak for the implementation of a EWS (more false positives than true positives). This is something that should be clearly accounted for. I suggest stating that this work represents just a starting point towards the implementation of a prototype EWS.

5- Just an advice: in the title and in the text you use the term "intensity-duration curve". This is correct, however in the international community the term "threshold" is more used than "curve". Your choice is correct but it could penalize your work in on-line search engines and databases indexing.

-

SPECIFIC COMMENTS

P1L11 (and elsewhere in the manuscript). If I have understood correctly, your methodology cannot be used for a EWS, because it doesn't use precipitation forecasts. Instead, there are two possibilities: either you state that your methodology can be used for nowcasting of landslide hazard in near-real time, or you state that your methodology represents a prototype version of a EWS that could be implemented in the (near?) future. The back-analyses you performed tested the potentiality of the prototype for future EWS applications. Please, rephrase the text where appropriate if my comment is correct, otherwise please describe better the possibility of providing forecasts for a EWS.

P1L19 The second type results. . .

[Figure]

P2L9 Please, remove blank line

P2L10-20. You use rainfall data from two different satellite missions (TRMM and GPM), which use different sensors. In general, such circumstances should be analyzed carefully because different sensors may produce slightly different measurements. Can you be sure that the precipitation values coming from these two different datasets are consistent?

P2L19-22. I suggest cutting this part (and a similar part which is found later in the manuscript). I think that the value of a research paper is to propose alternative approaches that have not been experienced before, therefore in my opinion the fact that you don't propose a "classic" ID threshold is not a drawback of your manuscript. On the contrary, it potentially makes your work more original and interesting. Concerning antecedent rainfall, I share your opinion of poor constraints with debris flow triggering. I suggest moving this sentence in the state of the art review. You can write that despite many recent works on rainfall thresholds took advantage of the use of antecedent rainfall or other hydrologic constraints (e.g. Posner et al., 2015; Bogaard et al., 2018; Segoni et al., 2018b), there is a general agreement that in case of debris flows on granular terrain (with relatively high hydraulic conductivity), the triggering time is well correlated with peak intensity and duration of the triggering rainfall (Caine, 1980; Guzzetti et al., 2008; Chen et al., 2015).

Page 3, Figure 2. I suggest deleting this figure. A figure based on such a large area is not needed for your work. It would be better to add a zoom on the study area as a third panel of figure 1.

Page 6, Figure 4, step II, Block 2, 2nd box: please, change "pixe" with "pixels".

Page 7, lines 17-18. "the threshold. . . the ID curve". This sentence is not clear and I think it is redundant with the previous one. I suggest deleting.

Page 8, Table 1. Please remove the blank row.

Page 8, Table 1. Something in this table is not clear to me. When you have a pair of Google Earth images bracketing two or more storms, how can you assess which storm triggered the landslides? In the text you explain that the landslides are given to the larger storm (lines 8-9). However, from the table it seems that landslides are shared among all the storms (see e.g. entries 2-3 and 4-5-6).

P9L5. How did you extract this curve? Is it just a subjective manual sketch to low-bound the experimental data or did you use some more robust approach? In case you use a subjective visual approach, you should comment in the discussion or conclusion that this is a weakness of the methodology and that this should be another improvement to carry out during the next phases of the research, when more data will be available.

P9L5. Please, write the equation of the curve.

P10Fig5. Please, add in the legend a key explaining the meaning of the dots. Add in the figure the threshold equation. Remove these information from the caption.

P10L8. There is no need to recap what you obtained in the previous chapter. If you want to introduce the section, you could briefly state what is the objective of the next steps.

P13L9. Instead of colors, please use class names that are related with the physical meaning of the classes.

P14L4-6. In my opinion this part is interesting and I would like to have more details, to evaluate quantitatively the outcomes. Can you provide quantitative statistics? E.g. the percentage of pixels $>3\sigma$, $>2\sigma$, and $>1\sigma$ that you found outside landslides and inside landslides.

P14L5. If you find a single pixel with these characteristics, is it enough to define a landslide? Or do you need to find a cluster of pixels? Please, clarify.

P14L6. These landslides were not "predicted". I would rephrase with "identified with the proposed method".

P15L7. Maybe the correct reference is to Fig. 3a.

P16L3. Please delete "/line".

Section 2.5. A sounder mathematical approach would help understanding the methodology. How were the A and B curves defined? I guess you translated upward the general Faifa curve to fit experimental data. Is that correct? Please, provide explanations.

Figure 9. Please, modify the figure to make it as much self-explaining as possible. E.g., write the curve equations, provide a key in the legend to explain the meaning of the colors of the points.

Section 2.6. Is it possible to show a figure depicting the pixels with correctly identified landslides (true positives) and the pixel forecasted as unstable (further subdivided into true positives and false positives)?

P19L3. Maybe "limited size" is more appropriate. By the way: does a landslide smaller than 25m represent a significant hazard in your study area?

P19L10-23. Usually a research describes what has been done and what will be done in the future should be summarized in a few lines. I suggest deleting the part dealing with future work. At least, reduce it consistently. In addition, most of these sentences could be moved in the state of the art description in the introduction (see some of my previous comments).

-

REFERENCES

Bogaard, T. and Greco, R.: Invited perspectives: Hydrological perspectives on precipitation intensity-duration thresholds for landslide initiation: proposing hydrometeorological thresholds, Nat. Hazards Earth Syst. Sci., 18, 31–39, 2018.

Caine, N. (1980). The rainfall intensity-duration control of shallow landslides and debris

flows. Geografiska annaler: series A, physical geography, 62(1-2), 23-27.

Cannon, S.H., Boldt, E.M.,Laber,J.L., Kean, J.W.,Staley,D.M., 2011.Rainfallintensity–duration thresholds for postfire debris-flow emergency-response planning. Nat. Hazards 59, 209–236.

Guzzetti, F., Peruccacci, S., Rossi, M., & Stark, C. P. (2007). Rainfall thresholds for the initiation of landslides in central and southern Europe. Meteorology and atmospheric physics, 98(3-4), 239-267.

Guzzetti, F., Peruccacci, S., Rossi, M., & Stark, C. P. (2008). The rainfall intensity–duration control of shallow landslides and debris flows: an update. Landslides, 5(1), 3-17.

Jakob, M., Owen, T., Simpson,T., 2012. A regional real-time debris flow warning system for the District of North Vancouver, Canada. Landslides 9(2), 165–178.

Lagomarsino, D., Segoni, S., Rosi, A., Rossi, G., Battistini, A., Catani, F., & Casagli, N. (2015). Quantitative comparison between two different methodologies to define rainfall thresholds for landslide forecasting. Natural Hazards and Earth System Sciences, 15(10), 2413-2423.

Posner, A. J. and Georgakakos, K. P.: Soil moisture and precipitation thresholds for realtime landslide prediction in El Salvador, Landslides, 12, 1179–1196, 2015.

Segoni, S., Piciullo, L., and Gariano, S. L.: A review of the recent literature on rainfall thresholds for landslide occurrence, Landslides, 15, 1483–1501, 2018a.

Segoni, S., Rosi, A., Lagomarsino, D., Fanti, R., and Casagli, N.: Brief communication: Using averaged soil moisture estimates to improve the performances of a regional-scale landslide early warning system, Nat. Hazards Earth Syst. Sci., 18, 807–812, 2018b.

2018-282, 2018.

---

## Referee Comment (RC2) · Anonymous Referee #2 · 19 Dec 2018

[referee-annotated manuscript omitted]

---

## Author Comment (AC1) · 25 Feb 2019

Please find the attached supplemental documents. 1. Cover Letter 2. Revised Manuscript 3. Responses to all the comments

Please also note the supplement to this comment:
https://www.nat-hazards-earth-syst-sci-discuss.net/nhess-2018-282/nhess-2018-282-AC1-supplement.zip

---

## Referee Report (RR1)

[referee-annotated manuscript omitted]